# Machine learning-based identification of contrast-enhancement phase of computed tomography scans

Siddharth Guha[1], Abdalla Ibrahim[1], Qian Wu[1], Pengfei Geng[1], Yen Chou[1], Hao Yang[1], Jingchen Ma[1], Lin Lu[1], Delin Wang[2], Lawrence H. Schwartz[1], Chuan-miao Xie[2], Binsheng Zhao[1]*

**1** Department of Radiology, Columbia University Irving Medical Center, New York, NY, United States of America, **2** Sun Yat-Sen University Cancer Center, Guangzhou, China

* zhaob1@mskcc.org

**Data Availability Statement:** All relevant data are within the paper and its Supporting Information files.

## Abstract

Contrast-enhanced computed tomography scans (CECT) are routinely used in the evaluation of different clinical scenarios, including the detection and characterization of hepatocellular carcinoma (HCC). Quantitative medical image analysis has been an exponentially growing scientific field. A number of studies reported on the effects of variations in the contrast enhancement phase on the reproducibility of quantitative imaging features extracted from CT scans. The identification and labeling of phase enhancement is a time-consuming task, with a current need for an accurate automated labeling algorithm to identify the enhancement phase of CT scans. In this study, we investigated the ability of machine learning algorithms to label the phases in a dataset of 59 HCC patients scanned with a dynamic contrast-enhanced CT protocol. The ground truth labels were provided by expert radiologists. Regions of interest were defined within the aorta, the portal vein, and the liver. Mean density values were extracted from those regions of interest and used for machine learning modeling. Models were evaluated using accuracy, the area under the curve (AUC), and Matthew's correlation coefficient (MCC). We tested the algorithms on an external dataset (76 patients). Our results indicate that several supervised learning algorithms (logistic regression, random forest, etc.) performed similarly, and our developed algorithms can accurately classify the phase of contrast enhancement.

## 1. Introduction

Computed tomography (CT) is routinely used in evaluating liver pathology and provides high diagnostic value. In particular, multi-phase contrast-enhanced or dynamic CT images have been shown to have high sensitivity and specificity in diagnosing primary liver lesions, such as hepatocellular carcinoma (HCC) [1–5]. In fact, dynamic CT plays an essential role in the diagnosis, treatment, and response assessment of hepatocellular carcinoma based on the widely utilized Liver Imaging Reporting and Data System (LI-RADS) [6]. However, not only are there significant differences in HCC lesion detection amongst the various phases of dynamic CT [7,

**Funding:** B.Z. and L.S. received financial support from the National Institutes of Health through grant U01 CA225431 (https://reporter.nih.gov/search/V24b0F7_OU-Zdua-r0RKIQ/project details/10417115) The funders had no role in study design, data collection and analysis, decision to publish, or preparation of the manuscript.

**Competing interests:** The authors have declared that no competing interests exist.

8], but different phases also provide different diagnostic accuracy for different types of lesions [9, 10]. In addition, certain phase-specific findings can be highly informative, such as hyper-vascularity of HCC lesions in the arterial phase being predictive of malignancy [11]. Because the varying physiology of patients can greatly impact the rate at which contrast perfuses the liver [12, 13], there can be significant variability in obtaining sufficient and quality images of each phase using acquisition protocols standardized upon timing after contrast injection. Thus, there exists a need to provide a reliable identification tool of dynamic CT phases for quality assurance.

Dynamic CT scans also provide abundant non-visual information to characterize liver lesions. With the advent of high-throughput data mining and advances in medical imaging techniques, radiomics has emerged as a promising field for extracting and utilizing these quantitative imaging findings for various clinical purposes. For example, various radiomic features (RFs) of HCC tumors, such as diameter, shape, intensity, or texture, have been reported to predict prognosis accurately [14]. One of the challenges in using features extracted from dynamic CT images is the reproducibility of RFs across different phases of the same scan, or similarly across contrast-enhanced CT scans. Many RFs ranging from tumor diameter [15] to tumor density [16] have been shown to vary in measurement by as much as 15% across different phases. Moreover, the vast majority RFs are poorly reproducible in measurement amongst different phases [17]. This further reinforces the need for a reliable and accurate method of identifying the contrast enhancement phase of CT scans.

Previous studies investigated the potential of machine learning models to accomplish the task of automatic and accurate identification of phases on dynamic CT images [18–22]. However, drawbacks from these prior models include the requirement for large numbers of annotated scans to train the model, the use of computationally expensive convolutional neural networks (CNNs), and the difficulty in the implementation and interpretation of models for broader clinical use. There also exists a dearth of information on the performance of different types of models, leaving no way to compare model types.

Given these challenges, this study aims to compare various supervised learning models to propose an easily implementable, high-accuracy model for phase identification. This was accomplished by training and validating five different supervised learning models on three different types of organ density inputs, then externally validating the performance on an external dataset. All tested supervised learning models demonstrated high accuracy in phase identification and show great potential for developing a fully automated phase classification system that can be used for lesion characterization, response assessment, and many other clinical tasks.

## 2. Materials and methods

### 2.1. Patient data and study overview

Data were obtained from two cohorts A, B totaling 135 patients with 2391 scan timepoints analyzed retrospectively (see Figs 1 and 2 for details): [A] dynamic CT scans from 59 patients with 2314 scan timepoints; [B] multiphase CT scans from 76 patients with 77 scans total (one scan of a single phase selected for each patient with one patient with scans of two different phases selected). Scans from Cohort A were from patients diagnosed with HCC (pre-treatment) who were imaged at Sun Yat-sen University Cancer Center using a dynamic CT protocol that took approximately 40 images every 2 seconds following contrast injection. Scans from Cohort B were aggregated from patients diagnosed with metastatic colorectal cancer enrolled in the Amgen Phase III PRIME (Panitumumab Randomized Trial In Combination With Chemotherapy for Metastatic Colorectal Cancer to Determine Efficacy) Clinical Trial and from patients at CUIMC diagnosed with cancer of various pathology with metastases to the liver.

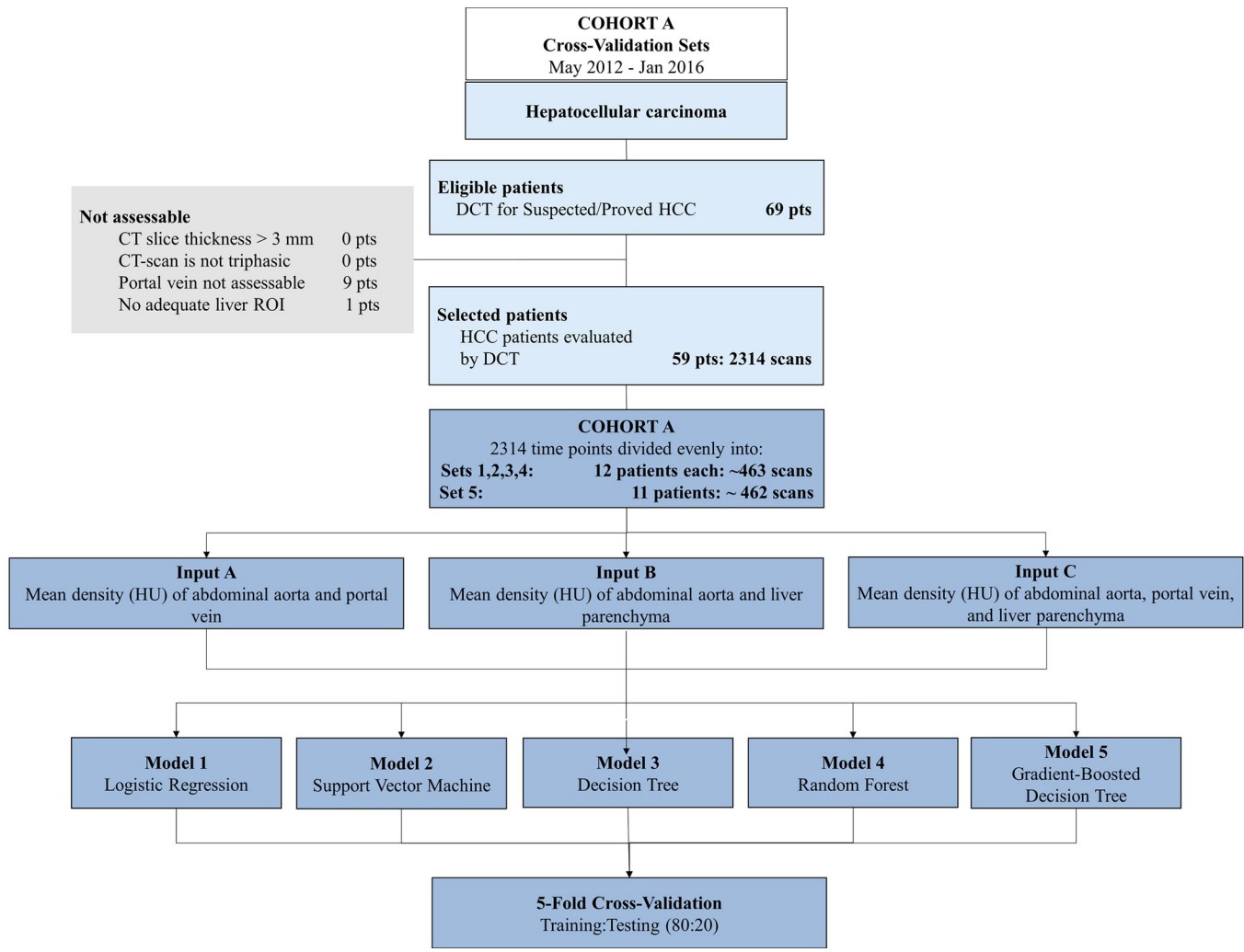

**Fig 1. Flow chart of cross-validation dataset.**

These scans were taken using a standard multiphase contrast-enhanced CT protocol where a single image was taken from a non-contrast enhanced phase, arterial phase, and portal venous phase. All scans were deidentified before collection.

In the first part of the study, multiple machine learning (ML) models were trained and tested to accomplish the target task of predicting the contrast-enhancement phases of images in the given datasets. In total, five different types of supervised learning models and three different combinations of organ density inputs were evaluated. In the first part of the study, k-fold cross-validation was performed using Cohort A to train and test each model with the two different inputs. The performance of each model was then compared between each of the two inputs to determine the better-performing input. It was also compared between models to determine the best-performing type of model.

In the second part of the study, the models were trained using the same inputs on data from Cohort A and then tested on Cohort B to evaluate the robustness of the models on external datasets with varying patient composition. Details of the differences in patient characteristics between the two datasets are summarized in Table 1.

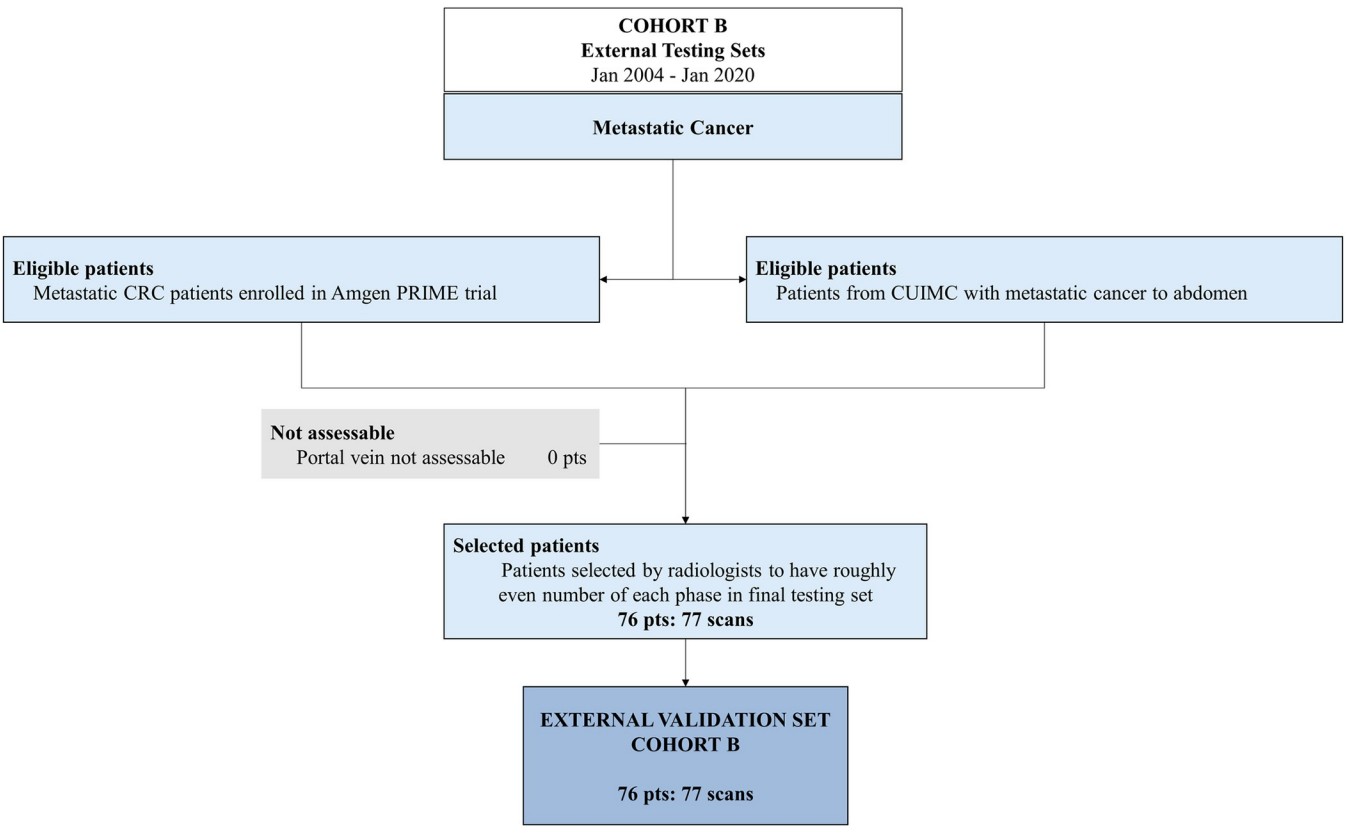

**Fig 2. Flow chart of external validation dataset.**

## 2.2. Dynamic CT image acquisition protocol

The dynamic CT images for Cohort A were obtained using a TOSHIBA Aquilon scanner at Sun Yat-sen University Cancer Center. The images were obtained using the following acquisition protocols: FC02, FC04, FL03 convolution Kernels, 8 mm slice interval, 0.56–1.0 mm pixel spacing, 500–14000 ms exposure time, 80 50–250 mA X-ray tube current, and 120 kVP. For each patient, scans were captured roughly every two seconds following contrast injection resulting in a total of 2314 images (on average 40 images per patient).

## 2.3. Reference standard for contrast-enhancement phases

Four phases of dynamic CT-scan images: (I) Non-contrast Enhanced Phase (NCE), (II) Early Arterial Phase (E-AP), (III) Late Arterial Phase (L-AP), and (IV) Portal Venous Phase (see Fig 3 for an example) were used as ground truth labels. Two radiologists (QW and PG, with four and five years of experience in abdominal imaging) independently assigned labels to each of the 2314 scan timepoints. The radiologists assigned the labels based on the LI-RADS Version 2018 criteria for defining dynamic CT phases [6] and other commonly used clinical criteria [23–25]. Disagreements over the labels were reviewed and discussed with a third radiologist (YW, with six years of experience), and a consensus was reached on the label. The scans in Cohort B consisted of scans with previously assigned labels, so roughly equal numbers of scans with each type of label were randomly chosen from a multi-center clinical trial dataset by our radiologists and provided as an external validation set. The patients in Cohort B were metastatic cancer patients with metastases to the abdomen and had a much wider range of

**Table 1. Patient characteristics.**

| | Cohort A | Cohort B |
|---|---|---|
| Training set | Yes | No |
| Cirrhosis | Yes | No |
| CT-scan acquisition | Dynamic CTs | Dynamic CTs |
| Histology | Liver nodule | Non-HCC |
| Number of patients | 59 | 76 |
| Age (years) | 52 ± 13 | / |
| Male | 49 (0.83) | / |
| Female | 10 (0.17) | / |
| Cirrhosis Cause Alcohol | 8 (.13) | / |
| Hepatitis B | 45 (.75) | / |
| Hepatitis C | 0 (.0) | / |
| HIV | 0 (.0) | / |
| Hemochromatosis | 0 (.0) | / |
| NASH | 0 (.0) | / |
| Unreported | 15 (.25) | / |
| CHILD score at diagnosis | | |
| A | 0 (.0) | / |
| B | 0 (.0) | / |
| C | 0 (.0) | / |
| Unreported | 59 (.100) | / |
| Pathology** | 42 (.70) | / |
| Well differentiated HCC | 2 (.03) | / |
| Well to moderately differentiated HCC | 5 (.08) | / |
| Moderately differentiated HCC | 19 (.32) | / |
| Moderately to poorly differentiated HCC | 12 (.20) | / |
| Poorly differentiated HCC | 4 (.07) | / |
| Unknown differentiation HCC | 0 (.0) | / |
| Non-HCC lesions | 4 (.07) | 76 (1.0) |
| -Malignant | 2 (.03) | 76 (1.0) |
| Cholangiocarcinoma*** | 1 (.02) | 1 (.01) |
| Metastasis | 1 (.02) | 75 (.99) |
| Benign | 2 (.03) | 2 (.03) |
| Hemangioma | 0 (.0) | 0 (.0) |
| FNH | 2 (.03) | 0 (.0) |
| Unreported | 14 (.23) | 0 (.0) |

Note.—*data are expressed in percentages. Etiology of cirrhosis are non-mutually exclusive. NASH = nonalcoholic steatohepatitis, HIV = Human Immunodeficiency Virus.

Note.—**Data are number of patients; data in parentheses are percentages for each group (HCC and Non–HCC).

*** including one case of hepatocholangiocarcinoma. FNH = focal nodular hyperplasia, / = not applicable.

pathology (including metastatic colorectal, pancreatic, lung, breast, and prostate cancer, to name a few) in order to increase the model's external validity. Information about the number of scans of each phase can be found in Table 2.

In a previous study [20], the labels for the reference standard were assigned using an algorithm based solely on quantitative thresholds for the mean density of the aorta and portal vein. However, upon further investigation, it was discovered that there are significant differences in the assignment of labels between the quantitative algorithm and radiologist visual inspection,

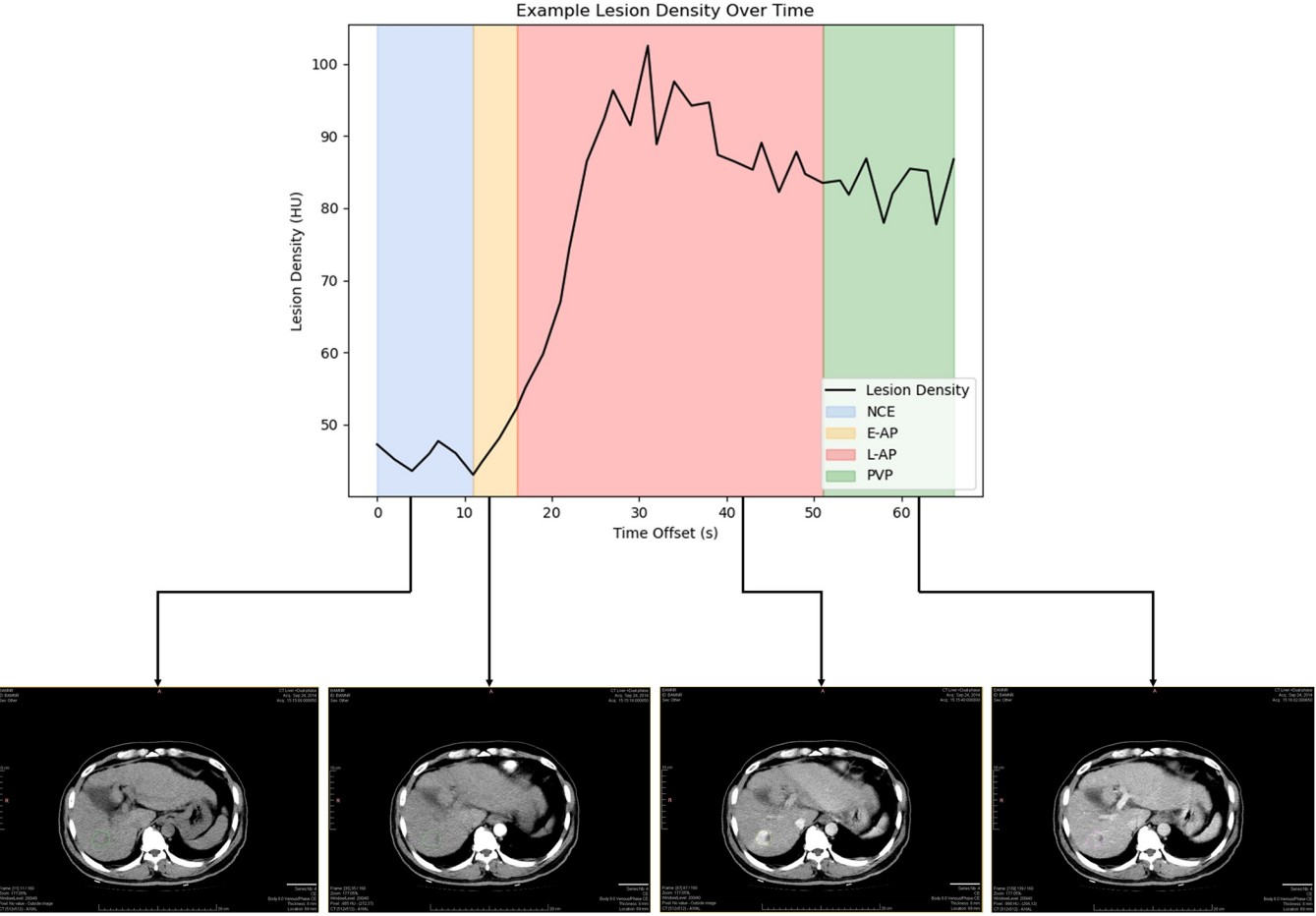

**Fig 3. Lesion density of HCC across contrast enhancement phases.**

with roughly only 80% agreement (Fig 4). Given these differences, cases were randomly selected for further review, and a consensus was reached that the radiologist-assigned labels were the most accurate based on the clinically used criteria (Fig 4).

## 2.4. Modeling and statistical analyses

For each CT image in each patient at each timepoint, regions of interest (ROIs) in various organs of about 2 cm in maximum diameter were manually drawn by radiologists blinded to assess the predicted outcome [20]. The 2 cm diameter ROIs of the liver were drawn such that they were entirely situated within the liver parenchyma without including any lesions or observable blood vessels. The mean density in Hounsfield Units (HU) for each ROI was then calculated. The three inputs used to train the models (A,B,C) differed in the regions of interest

**Table 2. Phase labels of scans.**

|  | Cohort A | Cohort B |
|---|---|---|
| NCE | 254 | 21 |
| E-AP | 144 | 16 |
| L-AP | 877 | 20 |
| PVP | 1039 | 20 |

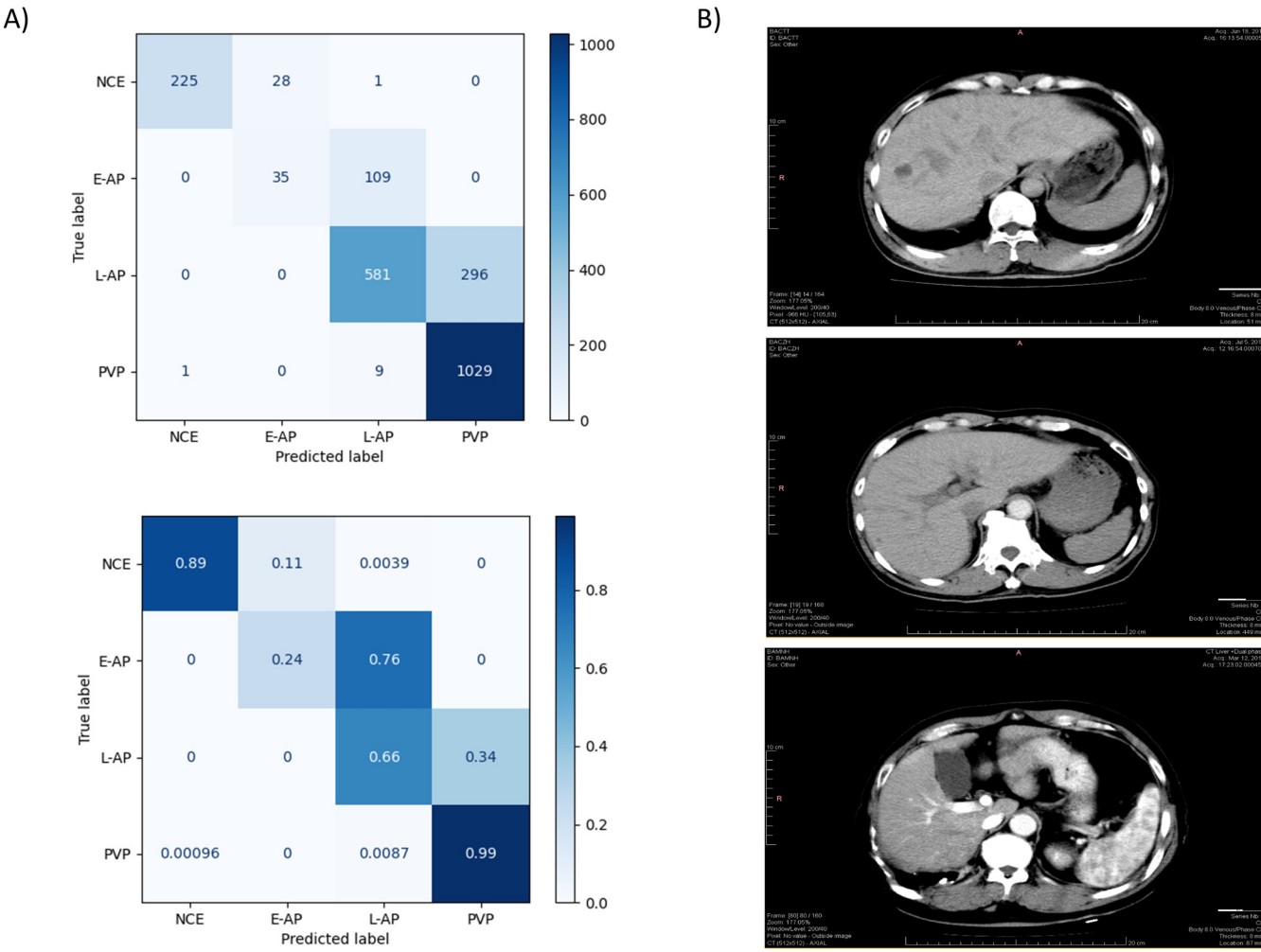

**Fig 4. A previously developed algorithmic approach to labeling dynamic CT phases has significant discrepancies with radiologist visual assessment.** (A) These confusion matrices (top = raw data, bottom = normalized) compare the "true" label (assigned by radiologist consensus) to the predicted label (determined by a previously developed quantitative algorithm). The algorithm shows significant inaccuracy in distinguishing early arterial phase (E-AP) from late arterial phase (L-AP) and late arterial phase (L-AP) from portal venous phase (PVP). (B) These scans are randomly selected examples of mislabeling by the algorithm. The top scan was labeled by the algorithm as E-AP and by radiologists as non-contrast-enhanced (NCE), the middle scan was labeled by the algorithm as L-AP and by radiologists as E-AP, and the bottom scan was labeled by the algorithm as PVP and by radiologists as L-AP.

(ROIs) from which the variables were obtained. Input A consisted of two variables–mean density of the abdominal aorta and portal vein. Input B consisted of two variables–mean density of the abdominal aorta and liver parenchyma. Input C consisted of three variables–mean density of the abdominal aorta, portal vein, and liver parenchyma. The abdominal aorta and portal vein were selected because a previous study identified these two ROIs as the most informative locations for predicting the contrast-enhancement phase [20]. The liver parenchyma was also included for comparison because it is easy to identify on scans, and its enhancement is of clinical interest in delineating lesions or other pathological abnormalities.

The output of the models was a categorical numerical value representing one of four contrast-enhancement phases. The consensus labels assigned by the radiologists of each CT image were converted into these values as follows: 0 = NCE, 1 = E-AP, 2 = L-AP, and 3 = PVP.

Given the nature of the phase classification task, the most appropriate model to use was a supervised learning model. Four supervised learning models (1,2,3,4,5) were selected based on

their frequent use in multi-label classification [26] and clinical applications [27–31]: (1) Logistic Regression (LR), (2) Support Vector Machine (SVM), (3) Decision Tree (DT), (4) Random Forest (RF), and (5) Gradient-Boosted Decision Tree (GBDT). All models were developed using standard libraries and packages in Python3.10.

To train and validate each model, a k-fold cross-validation approach was employed with k = 5, given that it achieves an optimal tradeoff between bias and variance [32–34]. The cross-validation was then repeated 6 times in order to obtain 30 total iterations of each model for robust statistical analysis. Each image from each timepoint of the dynamic CT scan of a single patient from Cohort A was treated as an independent scan timepoint from which the input of ROI densities was taken. To ensure each cross-validation set was independent, the scans were stratified by patient such that each set consisted of scans from unique patients. The logistic regression model was trained under default parameters except for a maximum number of iterations for convergence of 1000. The support vector machine model was trained under default parameters. The decision tree was trained using entropy at a maximum growing depth of 3. The random forest model was trained using 20 trees at a maximum growing depth of 8. The gradient-boosted decision tree model was trained using XGBoost (Extreme Gradient Boosting) with 20 trees at a maximum growing depth of 3.

Using the same five cross-validation training sets from Cohort A, Models 1–5 were trained on all three inputs, A, B, and C. The same ROI data was extracted for Cohort B, with each image acting as an independent scan timepoint. For each iteration of the training set, the entirety of Cohort B was used as the testing set to compare the models' predicted labels to the given labels.

All statistical analyses were performed in Python 3.10 using the scikit-learn library (version 1.0.2). In order to evaluate the performance of multiclass classification, the outputs needed to be binarized. Two approaches were used: One vs. Rest (OvR) and One vs. One (OvO). Given m distinct classes, the OvR method evaluates the model's ability to predict the given label vs. all other labels (m separate binary classifications). In contrast, the OvO method evaluates the model's ability to predict the given label vs. one other label, discarding the remainder of the dataset ((m)(m-1) separate binary classifications).

Precision-recall curves (PRCs), comparing the positive-predictive value (precision) to the sensitivity (recall), were used in lieu of receiver operating characteristic curves (ROCs) to evaluate model performance because ROCs can provide misleading information compared to PRCs on imbalanced datasets [35]. The accuracy, area under the PRC curve (AUPRC), and Matthews correlation coefficient (MCC) [36, 37] were calculated and averaged across all instances of the k-fold cross-validation. Furthermore, 95% confidence intervals were calculated for the accuracy and Matthews correlation coefficient. The MCC is calculated as shown:

$$MCC = \frac{TN \times TP - FN \times FP}{\sqrt{(TP + FP)(TP + FN)(TN + FP)(TN + FN)}}$$

TP = number of true positives, TN = number of true negatives, FP = number of false positives, FN = number of false negatives

## 3. Results

### 3.1. Phase classifier models: Training and validation in Cohort A

Each supervised learning model was trained and tested using an input of either mean density from Input A (aorta and portal vein), Input B (aorta and liver parenchyma), or Input C (aorta, portal vein, and liver parenchyma) with the 5-fold cross-validation approach outlined previously. An example of the normalized confusion matrices from a single instance of the cross-

## GBDT: Input C

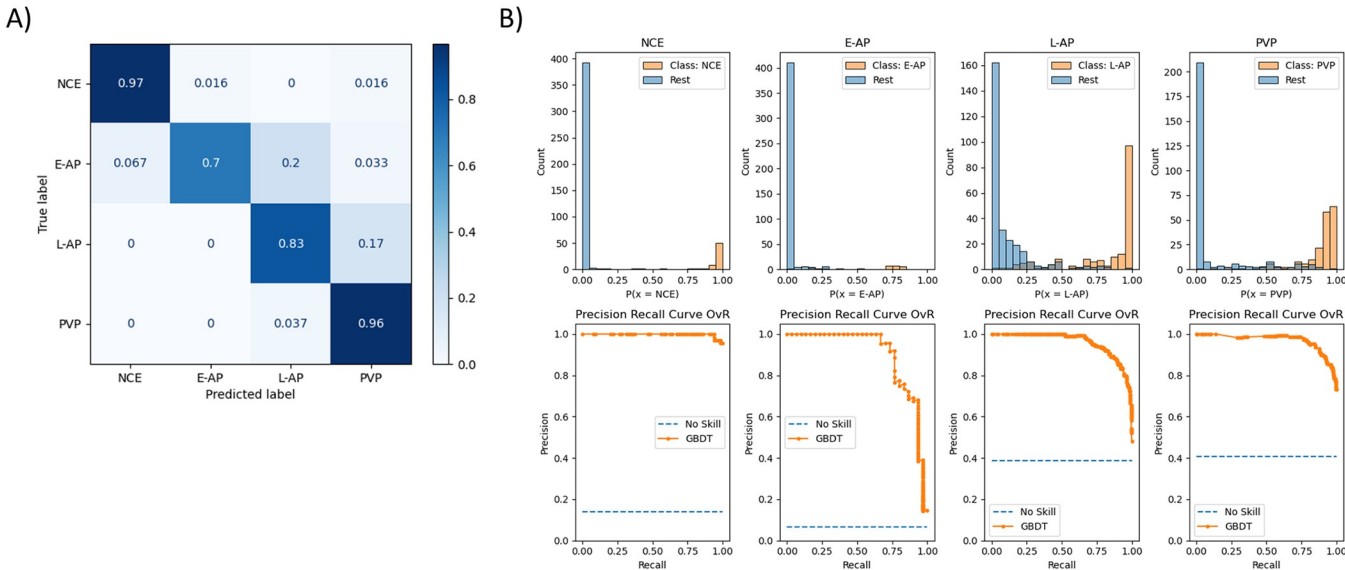

**Fig 5. Performance of the Gradient-Boosted Decision Tree (GBDT) model trained with Input C.** (A) displays the normalized confusion matrix. The greatest difficulty this model faced was in distinguishing early arterial phase (E-AP) from late arterial phase (L-AP) and late arterial phase (L-AP) from portal venous phase (PVP). (B) displays the precision-recall curves (PRC) using a One vs. Rest (OvR) approach. The OvR approach measures the ability of the model to distinguish that phase from all the other phases. For each pair of graphs, the top row displays histograms of the probability calculated by the model that a given scan is the target label. If that probability is greater than 0.5, then the model will classify it as the target label. Scans with the target label with probabilities greater than 0.5 and scans not of the target label ("Rest") with probabilities less than 0.5 are correctly classified. The bottom row of graphs in each pair displays the PRCs, which graph the model's precision (positive predictive value) against recall (sensitivity or true positive rate). A no-skill classifier is displayed as a horizontal line of the number of scans of the target label divided by the total number of scans. As shown by the PRC, the model has the most difficulty classifying E-AP scans.

validation for the GBDT model using Input C is shown in Fig 5A, and the PRCs and probability histograms are shown in Fig 5B.

When comparing models for a given input, all four models had similar overall performance in the classification of each scan (S1 Fig). In general, there was a very high accuracy in classifying NCE and PVP scans, with models achieving at least 90% prediction accuracy. However, all models had more difficulty in classifying E-AP and L-AP scans. Specifically, the models were prone to incorrectly labeling E-AP scans as L-AP as well as L-AP scans as PVP. There was not a significant difference in the ability to distinguish between these phases when models were trained with Input C as compared to Input A. Notably, models trained with Input B have a significant decrease in accuracy in identifying E-AP scans, with models correctly labeling such scans less than 40% of the time (and notably with 0% accuracy for the DT model) as compared to around 70% of the time for the other inputs.

The PRCs (S2–S4 Figs) reinforce that the models have little difficulty classifying NCE and PVP scans but struggle more with classifying E-AP and L-AP scans. All models trained exhibit a very good tradeoff of precision as recall increases for classifying NCE and PVP scans, but there is a sharper drop-off in the E-AP and L-AP curves towards a no-skill classifier, especially for the E-AP vs. NCE, E-AP vs. L-AP, L-AP vs. E-AP, and L-AP vs. PVP curves in the OvO analysis. This drop-off is even more pronounced for all models trained using Input B in the OvR E-AP curves and the OvO E-AP vs. L-AP and L-AP vs. E-AP (S3 Fig).

The accuracies, AUCs, and Mathews Correlation Coefficients (MCC) for all models are also marginally higher for Input C vs. the other two inputs but not in a statistically significant manner (Tables 3 and 4). Notably, the accuracy is significantly higher when calculated using the

**Table 3. The average accuracy, AUPRC, and MCC of the supervised learning models for the main dataset.**

| Model Type | Accuracy (Input A) | Accuracy (Input B) | Accuracy (Input C) | AUPRC (Input A) | AUPRC (Input B) | AUPRC (Input C) | MCC (Input A) | MCC (Input B) | MCC (Input C) |
|---|---|---|---|---|---|---|---|---|---|
| LR OvR | .9136 | .9147 | **.9356** | .8362 | .8150 | **.9012** | .7654 | .6960 | **.8051** |
| LR OvO | .8483 | .8326 | **.8860** | .8888 | .8695 | **.9306** | .6708 | .5552 | **.7185** |
| SVM OvR | .9315 | .9138 | **.9434** | .8873 | .8108 | **.9194** | .8061 | .6683 | **.8248** |
| SVM OvO | .8766 | .8173 | **.8942** | .9226 | .8666 | **.9424** | .7157 | .4950 | **.7378** |
| DT OvR | **.9223** | .9025 | .9215 | **.8870** | .8118 | .8836 | **.7809** | .5832 | .7785 |
| DT OvO | **.8700** | .7989 | .8688 | **.9226** | .8707 | .9211 | **.6990** | .3773 | .6972 |
| RF OvR | .9301 | .9157 | **.9448** | .8755 | .8179 | **.9146** | .7956 | .7054 | **.8281** |
| RF OvO | .8763 | .8355 | **.8984** | .9130 | .8701 | **.9375** | .7061 | .5642 | **.7424** |
| GBDT OvR | .9322 | .9167 | **.9461** | .8831 | .8255 | **.9206** | .8021 | .7073 | **.8302** |
| GBDT OvO | .8779 | .8353 | **.9008** | .9209 | .8751 | **.9438** | .7131 | 0.5656 | **.7487** |

OvR approach instead of the OvO approach. For example, the accuracies for models trained on Input C range from 92.15% to 94.61% for OvR and 86.88% to 90.08% for OvO. This is also corroborated by the non-overlapping 95% confidence intervals of the accuracies for the OvR vs. OvO models (Table 4). The highest average accuracy achieved was 94.61% by the GBDT model trained on Input C using the OvR analysis. Although most of the models perform more poorly on average on Input B and slightly better on average on Input C, these differences are not statistically significant for accuracy as shown by the overlapping confidence intervals. However, Inputs A and C have significantly higher MCC values as compared to Input B (Table 4). In addition, the DT model differed from all the other models in that its performance metrics were improved when trained on Input A vs. the other two inputs.

### 3.2. Phase classifier models: External validation in Cohort B

The same inputs were used from each of the five training sets in Cohort A, but this time the models were tested using scans from Cohort B. For comparison, an example of the normalized confusion matrices, PRCs, and probability histograms from a single instance of the cross-validation for the GBDT model using Input C are shown in Fig 6. As shown in the normalized confusion matrices and PRCs in S5–S8 Figs, the models accomplished the phase classification task with decreased but comparable accuracies. These accuracies are significantly lower than those in Cohort A in distinguishing between E-AP and L-AP, with most of the model-input

**Table 4. The 95% confidence intervals of accuracy and MCC of the supervised learning models for the main dataset.**

| Model Type | Accuracy (Input A) | Accuracy (Input B) | Accuracy (Input C) | MCC (Input A) | MCC (Input B) | MCC (Input C) |
|---|---|---|---|---|---|---|
| LR OvR | (.901, .926) | (.905, .925) | (.927, .944) | (.743, .788) | (.659, .733) | (.786, .824) |
| LR OvO | (.834, .863) | (.819, .846) | (.878, .894) | (.644, .698) | (.517, .594) | (.697, .740) |
| SVM OvR | (.922, .941) | (.904, .924) | (.936, .951) | (.787, .825) | (.624, .713) | (.807, .843) |
| SVM OvO | (.865, .888) | (.801, .834) | (.886, .902) | (.690, .741) | (.447, .543) | (.716, .760) |
| DT OvR | (.912, .933) | (.891, .914) | (.911, .932) | (.762, .800) | (.522, .644) | (.759, .798) |
| DT OvO | (.860, .880) | (.781, .816) | (.858, 879) | (.677, .721) | (.317, .438) | (.675, .719) |
| RF OvR | (.921, .939) | (.906, .926) | (.938, .952) | (.778, .814) | (.671, .739) | (.812, .844) |
| RF OvO | (.867, .886) | (.823, .848) | (.891, .905) | (.683, .729) | (.526, .602) | (.722, .762) |
| GBDT OvR | (.923, .941) | (.907, .927) | (.940, .953) | (.785, .819) | (.673, .741) | (.814, .846) |
| GBDT OvO | (.868, .888) | (.822, .849) | (.894, .908) | (.690, .736) | (.527, .604) | (.729, .768) |

## GBDT: Input C

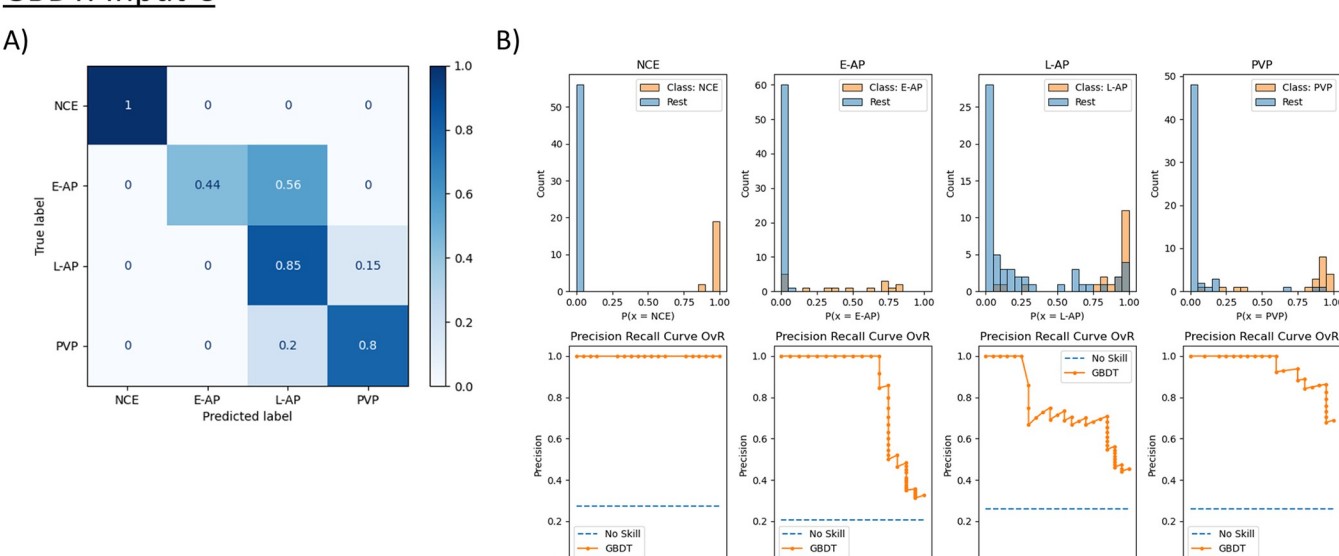

**Fig 6. Performance of the Gradient-Boosted Decision Tree (GBDT) model trained with Input C and tested on the external dataset.** (A) displays the normalized confusion matrix. The greatest difficulty this model faced was in distinguishing early arterial phase (E-AP) from late arterial phase (L-AP), even more so than in the original dataset. The model had comparable accuracy distinguishing late arterial phase (L-AP) from portal venous phase (PVP) to the original dataset. (B) displays the precision-recall curves (PRC) using a One vs. Rest (OvR) approach (See Fig 5 for details on interpretation). Similar to in the original dataset, the PRC shows that the model has the most difficulty classifying E-AP scans.

pairs achieving less than 50% accuracy for correctly classifying E-AP scans. This was especially highlighted in the models trained using Input B, where the SVM and DT models had 0% accuracy in correctly classifying E-AP scans, and the maximum accuracy was only 31% for the RF model.

However, when all performance metrics are averaged across all instances (Table 5), the performance is largely similar to that achieved in Cohort A. This is further supported by the overlapping or nearly overlapping confidence intervals of accuracy in the external dataset (Table 6). The greatest average accuracy achieved was 91.19% by the SVM model trained on Input A using the OvR analysis. One major difference in the performance in Cohort B is that

**Table 5. The average accuracy, AUPRC, and MCC of the supervised learning models for the external dataset.**

| Model Type | Accuracy (Input A) | Accuracy (Input B) | Accuracy (Input C) | AUC (Input A) | AUC (Input B) | AUC (Input C) | MCC (Input A) | MCC (Input B) | MCC (Input C) |
|---|---|---|---|---|---|---|---|---|---|
| LR OvR | **.8881** | .8738 | .8749 | **.9329** | .7993 | .8160 | **.7076** | .6425 | .6704 |
| LR OvO | **.7916** | .7464 | .7467 | **.9526** | .8703 | .8586 | **.5981** | .4925 | .5180 |
| SVM OvR | **.9119** | .8727 | .9095 | **.9459** | .8085 | .9201 | **.7735** | .6268 | .7672 |
| SVM OvO | **.8314** | .7445 | .8264 | **.9626** | .8760 | .9415 | **.6756** | .4464 | .6662 |
| DT OvR | **.9115** | .8766 | .9104 | **.8669** | .8613 | .8547 | **.7718** | .6241 | .7690 |
| DT OvO | **.8438** | .7560 | .8416 | **.9115** | .9106 | .9027 | **.6953** | .4544 | .6911 |
| RF OvR | **.9084** | .9071 | .9004 | **.9072** | .8653 | .9012 | **.7648** | .7548 | .7453 |
| RF OvO | **.8243** | .8196 | .8090 | **.9345** | .9099 | .9260 | **.6630** | .6548 | .6358 |
| GBDT OvR | **.9039** | .9019 | .8989 | **.9119** | .8607 | .8963 | **.7519** | .7399 | .7410 |
| GBDT OvO | **.8152** | .8056 | .8083 | **.9385** | .9062 | .9267 | **.6434** | .6243 | .6297 |

**Table 6. The 95% confidence intervals of accuracy and MCC of the supervised learning models for the external dataset.**

| Model Type | Accuracy (Input A) | Accuracy (Input B) | Accuracy (Input C) | MCC (Input A) | MCC (Input B) | MCC (Input C) |
|---|---|---|---|---|---|---|
| LR OvR | (.875, .901) | (.856, .891) | (.859, 891) | (.674, .741) | (.590, .695) | (.629, .712) |
| LR OvO | (.775, .808) | (.720, .772) | (.725, .768) | (.566, .630) | (. 438, .547) | (.473, .562) |
| SVM OvR | (.900, .923) | (.854, .891) | (.898, .921) | (.746, .801) | (.566, .688) | (.739, .796) |
| SVM OvO | (.817, .845) | (.717, .772) | (.812, .841) | (.648, .702) | (.383, .510) | (.638, .694) |
| DT OvR | (.901, .922) | (.859, .894) | (.900, .921) | (.745, .797) | (.561, .687) | (.744, .794) |
| DT OvO | (.833, .855) | (.729, .783) | (.831, .852) | (.674, .717) | (.389, .519) | (.669, .713) |
| RF OvR | (.896, .921) | (.894, .920) | (.887, .914) | (.736, .794) | (.720, .789) | (.713, .778) |
| RF OvO | (.809, .839) | (.801, .838) | (.792, .826) | (.634, .692) | (.619, 690) | (.604, .668) |
| GBDT OvR | (.891, .916) | (.888, .916) | (.886, .912) | (.722, .782) | (.702, .778) | (.711, .771) |
| GBDT OvO | (.800, .831) | (.786, 826) | (.793, .823) | (.613, .674) | (.586, .663) | (.600, .660) |

the models have a very slight improvement in performance when trained on Input A compared to Input C, which yielded the best results in Cohort A (Table 3). Similarly to testing in Cohort A, there is a relative weakness of models trained on Input B compared to Inputs A or C but only for the SVM and DT models, as shown by the non-overlapping confidence intervals for accuracy and MCC (Table 6). The accuracies, AUPRCs, and MCC scores for the LR, RF, and GBDT models trained on Input B are significantly unchanged compared to models trained on the other two inputs.

## 3.3. Phase classifier decision trees

The simple decision tree model was included for analysis because of its ease of construction and interpretability. Fig 7 shows the decision trees constructed for each type of input. The trees for Inputs A and C are almost identical, only differing in the cutoff used in the rightmost node at a depth of 2—aorta density less than 207.873 HU vs. liver density less than 119.996 HU for Input A vs. Input C, respectively. However, the branching at this node does not change the ultimate label received at the given maximum depth of three. Therefore, for all practical purposes, the two trees are identical. In contrast to the trees for Input A and C, the tree for Input B not only lacks a label for E-AP scans but also would have the same classification outcome at a maximum depth of 2 instead of 3.

## 4. Discussion

In order to optimize the automatic classification of dynamic CT scan images into the correct contrast-enhancement phases, we trained, tested, and externally validated five machine learning models using density information from three different ROIs. There were no significant differences in the performance of the various models, and the algorithms achieved relatively high accuracies around the range of 80–90%. The MCC scores of the models when tested in both the original and external datasets also suggest a strong positive correlation between the selected organ density measurements and the contrast-enhancement phase (since the MCC is a contingency matrix method of calculating the Pearson product-moment correlation coefficient [36], the interpretation of the statistic is similar). Thus, there is significant confidence that these models can be generalizable to the broader application of identifying phases during routine dynamic CT imaging.

When considering implementation for routine clinical use, the models have a significant advantage for two major reasons. First, the models incorporate only density measurements from the abdominal aorta, portal vein, or liver parenchyma taken from the standard of care

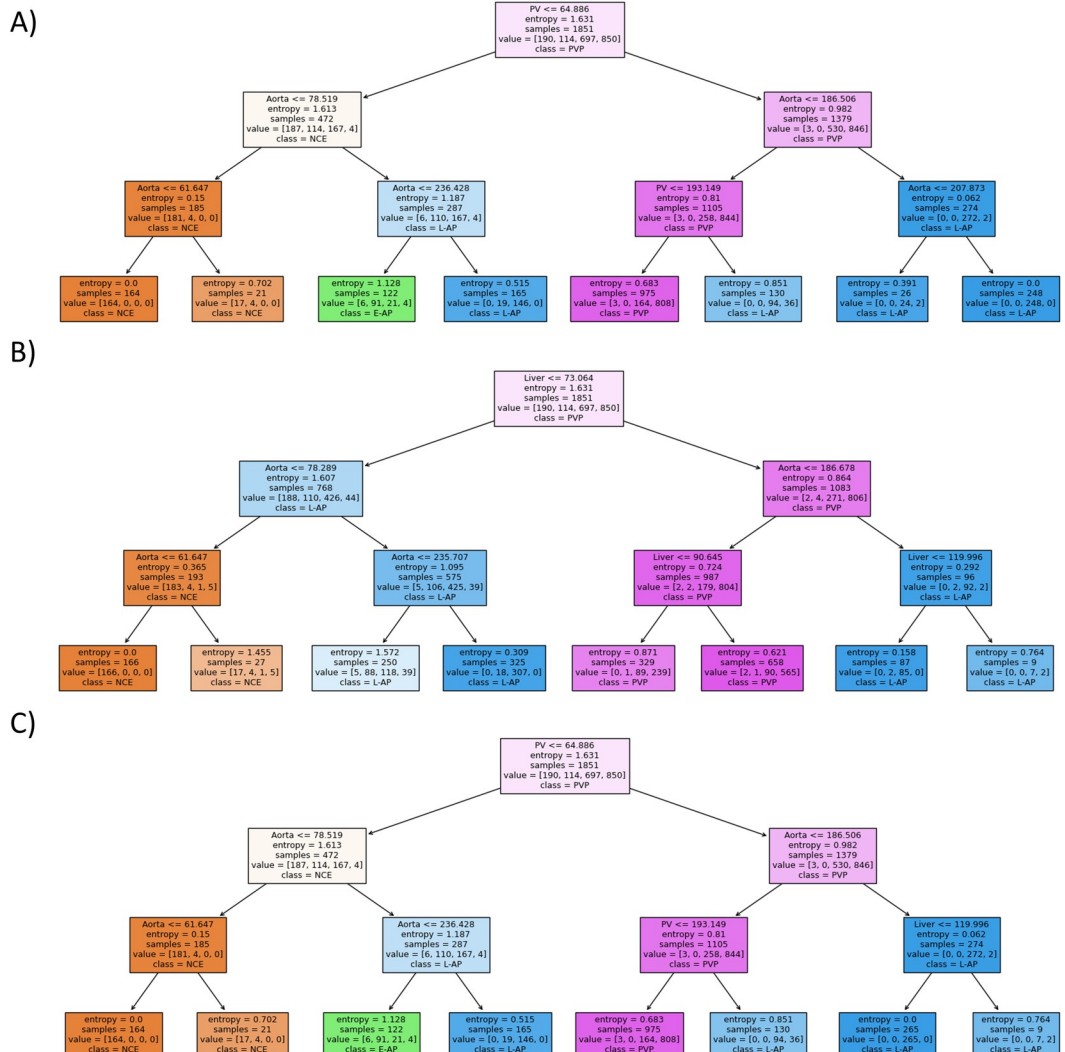

**Fig 7. Decision trees trained on the three inputs use similar cutoffs when splitting nodes.** (A), (B), and (C) display the decision trees constructed when trained on Inputs A, B, and C, respectively, from one of the cross-validation sets. Each node displays the cutoff for mean density of the given ROI that results in branching of the node. The terminal leaf nodes are the labels which are applied to the scans. Note that the tree constructed using Input B does not classify any scans as E-AP.

CT images, and this data is readily available and easily obtained. Second, the models tested were supervised learning models that are more easily interpretable than CNNs or deep learning models and more computationally efficient. In fact, the performance of each of our supervised learning models is highly comparable to models using multilayer CNNs [19, 38] or even generative adversarial networks (GANs) [21] trained on larger numbers of scans. Given these advantages, a simple supervised learning model such as logistic regression could be utilized for a fully automated phase classifier integrated into an imaging platform.

Part of the challenge in developing a fully automated phase classifier using our models is the automation of ROI detection. In our workflow, ROIs were manually drawn in the target organs for mean density measurement extraction, which can be a time-consuming process for a large number of scans. Luckily, there has been significant progress in automatically detecting the organs needed for the input. The abdominal aorta is clearly distinguishable on nearly all

dynamic CT images, and its automated segmentation has been accomplished with great accuracy and specificity [39–41]. The liver has also been a popular target of automatic segmentation, which has been accomplished by several groups, including our own [42–46]. Although it would be simple to use ROIs from solely these two organs, our results show that this input has lower accuracy in phase detection, especially for E-AP scans (S1 Fig), and lower correlation to phase as compared to using the aorta and portal vein or all three organs in conjunction (Table 4). In fact, the similarity of model performance between these latter two inputs and the construction of identical decision trees relying primarily on the aorta and portal vein ROIs suggests that the liver density may not be an informative or necessary input in the prediction of the contrast-enhancement phase. This could be due to the fact that the radiographic features of the liver parenchyma can be quite heterogeneous as a result of the pathophysiologic changes underlying cirrhosis or HCC and thus have too much variation to be reliably used, especially when compared to the liver parenchyma of patients with metastatic cancer to the liver. In addition, the dual blood supply of the liver from the hepatic artery and portal vein adds another element of variability to contrast enhancement and thus the reproducibility of any radiomic features that are sensitive to contrast, such as mean density. As a result, there is a need for automatic identification and segmentation of the portal vein on CT images. Although this is a significantly more challenging task, various groups (including our own) have made progress on that front [18, 47]. Since there is no significant change in performance when using all three organs, the development of a fully automated approach should proceed using only inputs from the aorta and portal vein for simplicity. The integration of automatic ROI detection with our supervised learning models presents a promising opportunity for future investigation.

Another consideration to further simplify the task of phase classification would be to employ a simple algorithmic approach based on hard quantitative cutoffs as dictated by a decision tree. Given the decent accuracy and performance of a simple decision tree with a maximum depth of three, the tree itself could be used to define the various contrast enhancement phases based on the relative densities of the aorta and portal vein. However, before these explicit cutoffs are determined, a decision tree model should be tested on a much larger dataset and externally validated across several other datasets. Future studies could then compare the accuracy of phase classification using quantitative thresholds, ML models, and radiologist visual inspection to validate this simple algorithmic approach for clinical use.

Because the ultimate goal of phase identification in dynamic CT is aiding physicians in clinical decision-making, special consideration must be paid to whether or not the models can accomplish such goals. As shown in our results, all the models face challenges in distinguishing early and late arterial phases. Since machine learning algorithms are known to be affected by imbalances in the datasets they are trained on, this result is likely due to the low number of E-AP scans skewing the model against identifying that phase. However, making such a distinction is difficult even for experienced radiologists who specialize in abdominal imaging. An essential factor to consider is whether such a distinction may not even be clinically necessary. Many tumor response assessment criteria such as the Morphology, Attenuation, Size, and Structure (MASS) criteria [48], modified Response Evaluation Criteria In Solid Tumors (mRECIST) [49], and Choi criteria [50] typically rely on measurements taken within the arterial phase (without regard for early or late) or portal venous phase in general. Given this information, the models are well-equipped to provide accurate identification for tumor response assessment and similar clinical tasks that do not heavily depend on distinguishing early and late arterial phases. On the other hand, radiomic signatures are likely to be more sensitive to the contrast enhancement phase. Previous studies reported that models using RFs vary in performance across phases [51, 52], which makes phase identification a necessary quality assurance step in studies analyzing contrast-enhanced CT scans. The precise impact of the contrast

enhancement phase on RFs reproducibility would need to be explored further in future research.

## 5. Conclusions

Simple supervised learning models using only the density from two common anatomical landmarks as an input can accomplish the task of phase identification with high accuracy comparable to more complex models. In the future, these models can be used to develop a fully automated and readily employable phase classifier that can improve the quality of clinical decision-making based on imaging features.

## Supporting information

**S1 Fig. The accuracy of the supervised learning models is roughly similar when using Input A or Input C but is decreased using Input B.** (A), (B), (C), (D) and (E) display the normalized confusion matrices for the logistic regression (LR), support vector machine (SVM), decision tree (DT), random forest (RF), and gradient-boosted decision tree (GBDT) models, respectively. Each matrix shows the performance on the same selected instance of cross-validation for models trained with Inputs A, B, and C from left to right. The greatest difficulty the models faced was in distinguishing early arterial phase (E-AP) from late arterial phase (L-AP) and late arterial phase (L-AP) from portal venous phase (PVP). Models trained using Input B have significantly greater difficulty correctly identifying E-AP than the models trained on the other two inputs.
(PDF)

**S2 Fig. This figure shows the precision-recall curves (PRCs) for each supervised learning model trained with Input A.** (A), (B), (C), (D), and (E) display the PRCs for the logistic regression (LR), support vector machine (SVM), decision tree (DT), random forest (RF), and gradient-boosted decision tree (GBDT) models, respectively. For each model, the graphs evaluated using a One vs. Rest (OvR) approach are shown on the top and a One vs. One (OvO) approach are shown on the (note that only the OvO PRCs for consecutive phases are shown). For each pair of graphs, the top row displays histograms of the probability calculated by the model that a given scan is the target label. If that probability is greater than 0.5, then the model will classify it as the target label. Scans with the target label with probabilities greater than 0.5 and scans not of the target label ("Rest" or "Class 0") with probabilities less than 0.5 are correctly classified. The bottom row of graphs in each pair displays the PRCs. A no-skill classifier is displayed as a horizontal line of the number of scans of the target label divided by the total number of scans.
(PDF)

**S3 Fig. This figure shows the precision-recall curves (PRCs) for each supervised learning model trained with Input B.** (A), (B), (C), (D) and (E) display the PRCs for the logistic regression (LR), support vector machine (SVM), decision tree (DT), random forest (RF), and gradient-boosted decision tree (GBDT) models, respectively. For each model, the graphs evaluated using a One vs. Rest (OvR) approach are shown on the top and a One vs. One (OvO) approach are shown on the bottom (note that only the OvO PRCs for consecutive phases are shown). See S2 Fig for more details on their interpretation.
(PDF)

**S4 Fig. This figure shows the precision-recall curves (PRCs) for each supervised learning model trained with Input C.** (A), (B), (C), (D) and (E) display the PRCs for the logistic regression (LR), support vector machine (SVM), decision tree (DT), random forest (RF), and

gradient-boosted decision tree (GBDT) models, respectively. For each model, the graphs evaluated using a One vs. Rest (OvR) approach are shown on the top and a One vs. One (OvO) approach are shown on the bottom (note that only the OvO PRCs for consecutive phases are shown). See S2 Fig for more details on their interpretation.
(PDF)

**S5 Fig. All models demonstrate slightly lower but still high accuracy on data from an external dataset.** (A), (B), (C), (D) and (E) display the normalized confusion matrices for the logistic regression (LR), support vector machine (SVM), decision tree (DT), random forest (RF), and gradient-boosted decision tree (GBDT) models, respectively. Each matrix shows the performance on the same selected training-testing pair for models trained with Inputs A, B, and C from left to right. The models faced greater difficulty than in Cohort A in distinguishing early arterial phase (E-AP) from late arterial phase (L-AP), especially when trained with Input B.
(PDF)

**S6 Fig. This figure shows the precision-recall curves (PRCs) for each supervised learning model trained with Input A and tested on the external dataset.** (A), (B), (C), (D) and (E) display the PRCs for the logistic regression (LR), support vector machine (SVM), decision tree (DT), random forest (RF), and gradient-boosted decision tree (GBDT) models, respectively. For each model, the graphs evaluated using a One vs. Rest (OvR) approach are shown on the top and a One vs. One (OvO) approach are shown on the bottom (note that only the OvO PRCs for consecutive phases are shown). See S2 Fig for more details on their interpretation.
(PDF)

**S7 Fig. This figure shows the precision-recall curves (PRCs) for each supervised learning model trained with Input B and tested on the external dataset.** (A), (B), (C), (D) and (E) display the PRCs for the logistic regression (LR), support vector machine (SVM), decision tree (DT), random forest (RF), and gradient-boosted decision tree (GBDT) models, respectively. For each model, the graphs evaluated using a One vs. Rest (OvR) approach are shown on the top and a One vs. One (OvO) approach are shown on the bottom (note that only the OvO PRCs for consecutive phases are shown). See S2 Fig for more details on their interpretation.
(PDF)

**S8 Fig. This figure shows the precision-recall curves (PRCs) for each supervised learning model trained with Input C and tested on the external dataset.** (A), (B), (C), (D) and (E) display the PRCs for the logistic regression (LR), support vector machine (SVM), decision tree (DT), random forest (RF), and gradient-boosted decision tree (GBDT) models, respectively. For each model, the graphs evaluated using a One vs. Rest (OvR) approach are shown on the top and a One vs. One (OvO) approach are shown on the bottom (note that only the OvO PRCs for consecutive phases are shown). See S2 Fig for more details on their interpretation.
(PDF)

**S1 File. This dataset contains the assigned contrast-enhancement phase and extracted ROI density features for each scan.**
(XLSX)

## Author Contributions

**Conceptualization:** Siddharth Guha, Abdalla Ibrahim, Lawrence H. Schwartz, Chuan-miao Xie, Binsheng Zhao.

**Data curation:** Qian Wu, Pengfei Geng, Yen Chou, Hao Yang, Delin Wang, Chuan-miao Xie.

**Formal analysis:** Siddharth Guha, Abdalla Ibrahim, Jingchen Ma, Lin Lu.

**Funding acquisition:** Lawrence H. Schwartz, Binsheng Zhao.

**Investigation:** Siddharth Guha, Abdalla Ibrahim, Qian Wu, Pengfei Geng, Yen Chou, Hao Yang, Jingchen Ma, Lin Lu, Delin Wang, Lawrence H. Schwartz, Chuan-miao Xie, Binsheng Zhao.

**Methodology:** Siddharth Guha, Abdalla Ibrahim, Jingchen Ma, Lin Lu.

**Project administration:** Lawrence H. Schwartz, Chuan-miao Xie, Binsheng Zhao.

**Resources:** Hao Yang, Delin Wang, Lawrence H. Schwartz, Chuan-miao Xie, Binsheng Zhao.

**Software:** Siddharth Guha, Hao Yang.

**Supervision:** Lawrence H. Schwartz, Binsheng Zhao.

**Writing – original draft:** Siddharth Guha, Abdalla Ibrahim, Binsheng Zhao.

**Writing – review & editing:** Siddharth Guha, Abdalla Ibrahim, Qian Wu, Pengfei Geng, Yen Chou, Hao Yang, Jingchen Ma, Lin Lu, Delin Wang, Lawrence H. Schwartz, Chuan-miao Xie, Binsheng Zhao.

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
