## [Decision Letter · Decision Letter 0]

1 Aug 2023

PONE-D-23-10431Machine learning-based identification of contrast-enhancement phase of computed tomography scansPLOS ONE

Dear Dr. Zhao,

Thank you for submitting your manuscript to PLOS ONE. After careful consideration, we feel that it has merit but does not fully meet PLOS ONE’s publication criteria as it currently stands. Therefore, we invite you to submit a revised version of the manuscript that addresses the points raised during the review process.

We look forward to receiving your revised manuscript.

Kind regards,

Shuai Ren

Academic Editor

PLOS ONE

Journal Requirements:

3. You indicated that ethical approval was not necessary for your study. We understand that the framework for ethical oversight requirements for studies of this type may differ depending on the setting and we would appreciate some further clarification regarding your research. Could you please provide further details on why your study is exempt from the need for approval and confirmation from your institutional review board or research ethics committee (e.g., in the form of a letter or email correspondence) that ethics review was not necessary for this study? Please include a copy of the correspondence as an "Other" file.

Reviewers' comments:

Reviewer's Responses to Questions

**Comments to the Author**

1. Is the manuscript technically sound, and do the data support the conclusions?

Reviewer #1: Yes

Reviewer #2: Partly

2. Has the statistical analysis been performed appropriately and rigorously? 

Reviewer #1: Yes

Reviewer #2: Yes

3. Have the authors made all data underlying the findings in their manuscript fully available?

Reviewer #1: Yes

Reviewer #2: Yes

4. Is the manuscript presented in an intelligible fashion and written in standard English?

Reviewer #1: Yes

Reviewer #2: Yes

5. Review Comments to the Author

Reviewer #1: Authors used radiomic features and machine learning to identify 4 labels of phase enhancement. The manual task is a time-consuming where there is a demand in an accurate automated labeling algorithm to identify the enhancement phase of CT scans. From 5 different machine learning models with three combinations of input regions, the type A (aorta and portal vein) combination was the most reasonable performance.

The process of driving and finding the best model is very thoughtful and completed. I hope to see the next step of actual motivation of pipeline or process of automated labeling algorithm and its performance. This manuscript is the first process to go on the next step. In that reason, the motivation and interest of this manuscript fall short of the expectations.

major recommendation:

1. it would be beneficial to the readers if you have graphical examples of three choices of uniform locations by y different level of contrasts that derived radiomic factor. It can be more convincing to the readers.

2. the series label from the site has a form of contrast labeling, which may not be consistent across the site.

what will be the practical method in using series label (text), radiomic factors from uniform ROI, or combined two types of factors (text and radiomic factor)? If it is out of scope, it would be good to discuss in the manuscript.

minor

1. are there any differences in model performance in CT technical parameters in misclassification? If so, elaborate it in the discussion.

2. add the discussion of the insight where the radiomic factors from liver parenchyma did not improve mode performance in phase enhancement. This is an unexpected finding that the application of this machine learning model is probably for the automatic labeling of CT images from the subjects with HCC diseases.

Reviewer #2: This paper proposes a method for identifying the contrast phase in multi-phase CT scans. The key idea is to identify suitable ROIs in the aorta, portal vein and liver parenchyma, and make a determination of the phase based on the mean intensity within the ROI. This determination is made by using common machine learning models such as logistic regression, SVM, decision tree, random forest and gradient boosted decision tree. The results show reasonably good performance. The writing is mostly clear and well organized.

Weaknesses include:

1. No confidence intervals or statistical tests (such as t-test) to support the claims of one method/inputs better than another.

2. On Cohort A, 5 fold cross-validation is used and each scan is assumed to be an independent sample. However, since many scans belong to the same patient, this assumption is flawed and the train-validation splits should be done at the patient level not at the scan level.

3. How was cohort A selected? Were any scans excluded due to artifacts etc? Were the patients picked randomly? Same questions for cohort B

4. In Section 2.4, the authors refer to "mean density" multiple times -- are they referring to "mean intensity" in Hounsfield units?

5. How was the 2 cm max diameter ROI drawn on the liver parenchyma? Were abnormalities in the liver such as lesions or major blood vessels excluded? What about inter-rater variability -- if the location of the ROI were changed, how does the prediction change (in other words, how sensitive was the prediction to changes in the location of the ROI?)

6. No multi-label metrics were reported -- for example, the multi-label accuracy (rather than one-vs-rest or one-vs-one) should be reported to get a sense of how effective the proposed method is in real usage. Perhaps top-k accuracy would also be very useful (for example, with k =2 )

6. PLOS authors have the option to publish the peer review history of their article (what does this mean?). If published, this will include your full peer review and any attached files.

Reviewer #1: No

Reviewer #2: No

---

## [Author Response · Author response to Decision Letter 0]

20 Sep 2023

We thank the editor and reviewers for their feedback. We have modified the manuscript accordingly where possible and provide a detailed point-by-point response to the reviewers below. 

Reviewer #1: Authors used radiomic features and machine learning to identify 4 labels of phase enhancement. The manual task is a time-consuming where there is a demand in an accurate automated labeling algorithm to identify the enhancement phase of CT scans. From 5 different machine learning models with three combinations of input regions, the type A (aorta and portal vein) combination was the most reasonable performance. 

The process of driving and finding the best model is very thoughtful and completed. I hope to see the next step of actual motivation of pipeline or process of automated labeling algorithm and its performance. This manuscript is the first process to go on the next step. In that reason, the motivation and interest of this manuscript fall short of the expectations. 

major recommendation:

1. it would be beneficial to the readers if you have graphical examples of three choices of uniform locations by y different level of contrasts that derived radiomic factor. It can be more convincing to the readers. 

We thank the reviewer for the comment. Figure 3 has been added to the manuscript to illustrate the various contrast-enhancement phases. 

2. the series label from the site has a form of contrast labeling, which may not be consistent across the site.

what will be the practical method in using series label (text), radiomic factors from uniform ROI, or combined two types of factors (text and radiomic factor)? If it is out of scope, it would be good to discuss in the manuscript.

We thank the reviewer for the comment. The labels were assigned by the three radiologists participating in the study following the LI_RADS guidelines, and the phases were assigned a numerical value as a categorical variable. No text analysis was involved in this manuscript, and the aim was to provide an algorithm that follows acceptable guidelines and provide reproducible values/labels.

minor

1. are there any differences in model performance in CT technical parameters in misclassification ? If so, elaborate it in the discussion. 

We thank the reviewer for the comment. We analyzed the patient data in a longitudinal manner, and the scans of each patient were taken using the same parameters. Therefore, we anticipate that the variations in RFs’ values are due to the varying time of acquisition following contrast injection. Furthermore, the models were validated using the external dataset which had different CT technical parameters for image acquisition with similar performance.

2. add the discussion of the insight where the radiomic factors from liver parenchyma did not improve mode performance in phase enhancement. This is an unexpected finding that the application of this machine learning model is probably for the automatic labeling of CT images from the subjects with HCC diseases. 

We thank the reviewer for the comment. We have added some explanation in the discussion on the reasons why the liver parenchyma radiomic factors did not improve model performance for predicting imaging phase, including the underlying physiology of blood flow in cirrhosis and the dual supply of the liver by the hepatic artery and portal vein (please see Page 11, Lines 48-51).

Reviewer #2: This paper proposes a method for identifying the contrast phase in multi-phase CT scans. The key idea is to identify suitable ROIs in the aorta, portal vein and liver parenchyma, and make a determination of the phase based on the mean intensity within the ROI. This determination is made by using common machine learning models such as logistic regression, SVM, decision tree, random forest and gradient boosted decision tree. The results show reasonably good performance. The writing is mostly clear and well organized. 

Weaknesses include:

1. No confidence intervals or statistical tests (such as t-test) to support the claims of one method/inputs better than another.

We thank the reviewer for the suggestion. We repeated the analysis using 6 repetitions of the 5-fold-cross validation in order to have 30 iterations for robust statistical evaluation. We have added 95% confidence intervals for accuracy and MCC as two additional tables within the manuscript, Tables 4 and 6. Furthermore, the results and discussion within the manuscript have been updated to reflect these findings where applicable. 

2. On Cohort A, 5 fold cross-validation is used and each scan is assumed to be an independent sample. However, since many scans belong to the same patient, this assumption is flawed and the train-validation splits should be done at the patient level not at the scan level.

We thank the reviewer for bringing this to our attention. The train-validation splits were done on patient-level, i.e lesions from the same patient would all be either in the training or the validation sets. We updated the methods section to clarify this (please see Page 6, Lines 19-21). 

3. How was cohort A selected? Were any scans excluded due to artifacts etc? Were the patients picked randomly? Same questions for cohort B

We thank the reviewer for the comment. The inclusion criteria for the two cohorts are included in the flowcharts in Figures 1 and 2. 

4. In Section 2.4, the authors refer to "mean density" multiple times -- are they referring to "mean intensity" in Hounsfield units?

We thank the reviewer for the comment. We are indeed referring to the mean intensity in Hounsfield units with the term density since the typical radiological convention is to refer to lesions on CT scans in terms of density. 

5. How was the 2 cm max diameter ROI drawn on the liver parenchyma? Were abnormalities in the liver such as lesions or major blood vessels excluded? What about inter-rater variability -- if the location of the ROI were changed, how does the prediction change (in other words, how sensitive was the prediction to changes in the location of the ROI?)

We thank the reviewer for the comment. The 2 cm diameter ROIs of the liver were drawn such that they were entirely situated within the liver parenchyma without including any lesions or observable blood vessels. This description was added to the methods section (please see Page 5, Lines 37-39). The variability of the liver ROI location was not studied because in general the liver ROI did not improve model performance, but it is a possible idea for future study. The portal vein and aorta ROIs are fairly homogenous since they are blood-filled, so as long as the ROIs still remain within the vessels it is unlikely to have a significant impact on the model.

6. No multi-label metrics were reported -- for example, the multi-label accuracy (rather than one-vs-rest or one-vs-one) should be reported to get a sense of how effective the proposed method is in real usage. Perhaps top-k accuracy would also be very useful (for example, with k =2 ) 

We thank the reviewer for the comment. The utilization of multilabel classification techniques such as one-vs-rest or one-vs-one is rooted in the nature of specific algorithms such as logistic regression and SVM. These algorithms inherently seek to establish linear separations (in other words, solve binary classification problems) within an N-dimensional space. Thus, when confronted with multilabel classification tasks, it is necessary to transform the problem into several binary classification sub-problems through a one-vs-rest or one-vs-one approach. Other multilabel classification algorithms such as top-k-accuracy would not be directly applicable to logistic regression and SVM and so the former techniques were chosen to allow for comparison amongst models. Furthermore, the one-vs-rest or one-vs-one approach are widely accepted techniques for obtaining multi-label metrics.

---

## [Decision Letter · Decision Letter 1]

6 Nov 2023

Machine learning-based identification of contrast-enhancement phase of computed tomography scans

PONE-D-23-10431R1

Dear Dr. Zhao,

We’re pleased to inform you that your manuscript has been judged scientifically suitable for publication and will be formally accepted for publication once it meets all outstanding technical requirements.

Kind regards,

Shuai Ren

Academic Editor

PLOS ONE

Additional Editor Comments (optional):

Dear authors, please incorporate the comments raised by the reviewer # 2 when you check the proof.

Reviewers' comments:

Reviewer's Responses to Questions

**Comments to the Author**

1. If the authors have adequately addressed your comments raised in a previous round of review and you feel that this manuscript is now acceptable for publication, you may indicate that here to bypass the “Comments to the Author” section, enter your conflict of interest statement in the “Confidential to Editor” section, and submit your "Accept" recommendation.

Reviewer #2: All comments have been addressed

2. Is the manuscript technically sound, and do the data support the conclusions?

Reviewer #2: Yes

3. Has the statistical analysis been performed appropriately and rigorously? 

Reviewer #2: Yes

4. Have the authors made all data underlying the findings in their manuscript fully available?

Reviewer #2: Yes

5. Is the manuscript presented in an intelligible fashion and written in standard English?

Reviewer #2: Yes

6. Review Comments to the Author

Reviewer #2: I thank the authors for addressing my comments. One minor comment: I think the authors should consider combining Tables 3 and 4 (and Tables 5 and 6) -- i think it is better to keep the metrics and the confidence intervals together in the same table.

7. PLOS authors have the option to publish the peer review history of their article (what does this mean?). If published, this will include your full peer review and any attached files.

Reviewer #2: No

---

## [Editor Report · Acceptance letter]

25 Jan 2024

PONE-D-23-10431R1 

PLOS ONE

Dear Dr. Zhao, 

I'm pleased to inform you that your manuscript has been deemed suitable for publication in PLOS ONE. Congratulations! Your manuscript is now being handed over to our production team.

Kind regards, 

on behalf of

Dr. Shuai Ren 

Academic Editor

PLOS ONE